# How ITD Insertion Sites Orchestrate the Biology and Disease of FLT3-ITD-Mutated Acute Myeloid Leukemia

**DOI:** 10.3390/cancers15112991

**Published:** 2023-05-30

**Authors:** Tobias R. Haage, Burkhart Schraven, Dimitrios Mougiakakos, Thomas Fischer

**Affiliations:** 1Department of Hematology and Oncology, Medical Center, Otto-von-Guericke University, 39120 Magdeburg, Germany; 2Gesundheitscampus Immunology, Inflammation and Infectiology (GC-I3), Medical Center, Otto-von-Guericke University, 39120 Magdeburg, Germany; 3Institute for Molecular and Clinical Immunology, Medical Faculty, Otto-von-Guericke University, 39120 Magdeburg, Germany; 4Center of Health and Medical Prevention (CHaMP), Otto-von-Guericke University, 39106 Magdeburg, Germany

**Keywords:** acute myeloid leukemia (AML), FLT3 mutation, internal tandem duplications (FLT3-ITD), non-juxtamembrane domain (non-JMD) insertion site, tyrosine kinase inhibition (TKI), chemotherapy resistance

## Abstract

**Simple Summary:**

*FLT3* gene mutations are among the most common genetic aberrations detected in AML and occur with a frequency of approximately 30%, mainly as internal tandem duplications (FLT3-ITD). As a novel finding, it has been reported that the specific insertion sites (IS) of FLT3-ITD exhibit marked heterogeneity in both biological and clinical features. Thus, the so-called non-juxtamembrane domain (non-JMD) FLT3-ITD insertions have been shown to be associated with worse clinical outcomes and resistance to both chemotherapy and tyrosine kinase inhibition. This present review summarizes our current knowledge of the biological and clinical impact of FLT3-ITD inserting at the non-JMD level. Recent evidence suggests that conformational changes depending on FLT3-ITD localization affect downstream signaling networks and the oncogenic potential. We propose that refined risk stratification guidelines integrating the negative prognostic impact of non-JMD FLT3-ITD are warranted. Overcoming therapy resistance in non-JMD-inserting FLT3-ITD-mutated AML may lead to promising treatment approaches.

**Abstract:**

Mutations of the *FLT3* gene are among the most common genetic aberrations detected in AML and occur mainly as internal tandem duplications (FLT3-ITD). However, the specific sites of FLT3-ITD insertion within *FLT3* show marked heterogeneity regarding both biological and clinical features. In contrast to the common assumption that ITD insertion sites (IS) are restricted to the juxtamembrane domain (JMD) of FLT3, 30% of FLT3-ITD mutations insert at the non-JMD level, thereby integrating into various segments of the tyrosine kinase subdomain 1 (TKD1). ITDs inserted within TKD1 have been shown to be associated with inferior complete remission rates as well as shorter relapse-free and overall survival. Furthermore, resistance to chemotherapy and tyrosine kinase inhibition (TKI) is linked to non-JMD IS. Although FLT3-ITD mutations in general are already recognized as a negative prognostic marker in currently used risk stratification guidelines, the even worse prognostic impact of non-JMD-inserting FLT3-ITD has not yet been particularly considered. Recently, the molecular and biological assessment of TKI resistance highlighted the pivotal role of activated WEE1 kinase in non-JMD-inserting ITDs. Overcoming therapy resistance in non-JMD FLT3-ITD-mutated AML may lead to more effective genotype- and patient-specific treatment approaches.

## 1. Introduction

In recent decades, research on acute myeloid leukemia (AML) has increasingly focused on genetic aberrations to identify and monitor molecular markers and their prognostic significance [1]. These findings became an integral part of the classification and risk assessment of myeloid neoplasms and are of indispensable importance to current strategies for the diagnosis and treatment of AML patients. Among the first molecular markers identified were internal tandem duplications (ITDs) in the gene encoding fms-like tyrosine kinase-3 (*FLT3*). The receptor tyrosine kinase FLT3 is composed of an extracellular module with five immunoglobulin-like domains, a transmembrane domain, and an intracellular module consisting of a juxtamembrane domain (JMD) and two tyrosine kinase subdomains (TKD1 and TKD2) connected by a kinase insert domain (Figure 1) [2]. In 1996, Nakao et al. detected in-frame ITDs of varying lengths in the JMD of the *FLT3* gene in a small cohort of AML patients [3]. ITDs were shown to vary in length from three base pairs (bp) to more than 400 bp, expected to result in a functional protein due to their in-frame nature [4]. Furthermore, a more C-terminal-located insertion site (IS) has been shown to be associated with higher ITD lengths [5,6,7]. Different FLT3-ITD mutations may be present in one patient or may arise or change during the disease process [8,9]. *FLT3* gene mutations occur with a frequency of about 25–30% in AML [10,11], the vast majority as ITDs with varying IS (FLT3-ITD) or as tyrosine kinase domain (TKD) point mutations in about 5% (FLT3-TKD). FLT3-TKD point mutations most commonly result in the substitution of aspartic acid at amino acid position 835, referred to as D835 mutation, at the TKD2 level [12]. Thus, mutations affecting *FLT3* are among the most common genetic aberrations detected in AML.

Re-examining the common assumption that ITDs exclusively occur in the JMD of *FLT3*, Breitenbuecher et al. detected ITDs inserted at the non-JMD level in approximately 30% of FLT3-ITD-mutated AML patients [13]. Non-JMD FLT3-ITD has been shown to be associated with inferior rates of complete remission (CR), shorter relapse-free survival (RFS), and shorter overall survival (OS) [5,14,15]. Moreover, non-JMD-inserting FLT3-ITD is also associated with resistance to chemotherapy and tyrosine kinase inhibitors (TKI). This present review addresses the biological and clinical impact of FLT3-ITD with non-JMD IS. Furthermore, we illustrate their high-risk phenotype, the heterogeneity of clinical response, and the current knowledge of the underlying biology. This review aims to highlight the pivotal role of non-JMD FLT3-ITD as an independent adverse prognostic factor and to emphasize its diagnostic and therapeutic importance. Queries performed for the selection of references in accordance with the corresponding sections are listed in Table 1. 

## 2. Signaling of FLT3 and FLT3-ITD

FLT3 is a membrane-bound, ligand-activated class III receptor tyrosine kinase that is physiologically expressed on hematopoietic stem and progenitor cells of both the lymphoid and myeloid lineage [16]. Conformational changes following FLT3 ligand (FLT3L) binding result in receptor dimerization, autophosphorylation, and activation of downstream signaling pathways, including signal transducer and activator of transcription 5 (STAT5), phosphatidylinositol 3-kinase (PI3K)/AKT, and mitogen-activated protein kinase/extracellular signal-regulated kinase (MAPK/ERK) [17,18,19,20]. In the absence of external ligands, FLT3 assumes an inactive and autoinhibited conformation that is maintained by the JMD. In AML, the presence of ITDs structurally interferes with the JMD-mediated autoinhibition, allowing FLT3 to maintain or switch to its active conformation even in the absence of FLT3L binding [21]. Consistent with the physiological functions of FLT3 in cell proliferation, differentiation, and survival [16], FLT3-ITD mutations result in constitutive activation that provides biological advantages in AML [22]. Compared with FLT3-ITD, FLT3 wild-type and FLT3-TKD point mutations show only weak phosphorylation of STAT5 and thus weak activation of STAT5-dependent target genes [23,24]. FLT3-ITD-associated activation of STAT5 leads to growth-factor-independent proliferation, upregulation of reactive oxygen species (ROS)-formation, and ROS-dependent DNA damage [25,26,27,28]. Stronger phosphorylation signals were found in FLT3-ITD-mutated AML patients with higher ITD lengths [29]. Aberrant cell-death signaling through receptor-interacting serine/threonine-protein kinase 1 (RIPK1) and upregulation of anti-apoptotic myeloid cell leukemia-1 (MCL-1), aberrant cell-cycle regulation, among others through CDC25A and CDC25C, and aberrant oncogenic signaling through phosphorylation of cytokine receptor common subunit beta (CSF2RB) were detected in FLT3-ITD-mutated cells [30,31,32,33,34,35]. Furthermore, FLT3-ITD mediates metabolic effects leading to the upregulation of aerobic glycolysis and cell-extrinsic effects by affecting dendritic cells and T cells [36,37]. 

## 3. FLT3 Tyrosine Kinase Inhibition

Following the detection of FLT3 and especially FLT3-ITD as a molecular marker in AML, larger cohorts of AML patients were studied to determine its prognostic value, and first efforts were made to establish a specific and targeted therapy against FLT3. In accordance with these studies, the occurrence of FLT3-ITD is associated with leukocytosis, a high percentage of peripheral blood and bone marrow blast cells, an increased relapse risk (RR) along with reduced RFS, and worse OS [9,10,38,39,40,41,42]. To date, the United States Food and Drug Administration (FDA) approved two tyrosine kinase inhibitors (TKIs): Midostaurin as first-line treatment in combination with cytarabine and daunorubicin in *FLT3*-mutated AML [43,44,45], and gilteritinib as single-agent therapy in relapsed or refractory AML with *FLT3* gene mutations [46,47]. Although studies have also shown survival benefits for sorafenib [48] and quizartinib [49,50], these TKIs have not yet been approved for use in *FLT3*-mutated AML patients. Midostaurin and gilteritinib bind to the adenosine triphosphate (ATP)-binding site of FLT3 in an activated conformational state and are designated as type I inhibitors along with others, including sunitinib and crenolanib. Type II inhibitors such as sorafenib, ponatinib, and quizartinib bind to the ATP-binding site and to a region adjacent to it and only in an inactivated conformational state of FLT3 [51,52,53]. In addition to these different mechanisms of interaction with the FLT3 receptor, first-generation (including midostaurin and sorafenib) and next-generation TKIs (including gilteritinib and quizartinib) are distinguished depending on the specificity of FLT3 bind-ing and off-target effects. Besides binding to FLT3, first-generation TKIs also bind to other kinases resulting in higher off-target effects and toxicity, whereas next-generation TKIs inhibit FLT3 more specifically and more potently. Response to TKI may be compromised by primary inherent or secondary acquired resistance mechanisms, up to and including treatment failure [54,55]. For the latter, incomplete elimination of *FLT3*-mutated cells after first-line treatment may be of particular importance. In vitro, Ba/F3 cells retrovirally transduced with different *FLT3* mutations showed differential sensitivity to the TKI tandutinib [56]. Furthermore, clonal heterogeneity, compensatory survival signaling pathways, acquisition of additional genetic alterations within the *FLT3* gene, as well as increased stimulation from overexpression of FLT3L and FLT3 itself contribute to resistance and subsequently decreased TKI impact or chemotherapy susceptibility [55,57,58,59,60,61,62,63]. Cytokine-mediated effects by granulocyte-macrophage colony-stimulating factor (GM-CSF), interleukin-3 (IL-3), and others also contribute to therapy resistance of FLT3-ITD-mutated AML [64].

## 4. Prognostic Impact of FLT3-ITD Mutations and Current Risk Stratification

ITDs were shown to vary in length from 3 bp to more than 400 bp [5]. The prognostic impact of ITD length in FLT3-ITD-mutated AML has been controversial, with studies showing either significantly worse CR rates and OS with increasing ITD lengths [65], better outcomes with increasing ITD lengths [66], or even no differences [10,67]. Recently, a systematic review and meta-analysis showed that longer ITD lengths are associated with a moderately but significantly increased hazard ratio for death [68]. The negative prognostic value of FLT3-ITD mutations was already recognized in the early 2000s and described by Abu-Duhier et al. and Kottaridis et al., among others [39,69]. In 2002, Thiede et al. identified a high FLT3-ITD mutant-to-wild-type ratio > 0.78 to be associated with a shorter OS as an independent prognostic factor in AML [40]. Another study of 579 FLT3-ITD-mutated AML patients revealed that an FLT3-ITD allelic ratio > 0.5 was associated with worse CR rates and shorter OS, and an FLT3-ITD allelic ratio > 0.8 related to worse RFS [70]. In a small cohort of 118 patients with FLT3-ITD-mutated AML studied by Jentzsch et al., a high FLT3-ITD allelic ratio was consistent with a higher proportion of blast cells in the bone marrow and peripheral blood at diagnosis [71]. Beyond inferior RR and OS with increasing FLT3-ITD mutant burden, Gale et al. highlighted the beneficial impact of a concomitant *NPM1* mutation [10]. In the 2017 European LeukemiaNet (ELN) assessment of molecular cytogenetic risk groups in AML, FLT3-ITD was implicated in risk stratification depending on its allelic ratio and the *NPM1* mutation status [72,73,74]. A high FLT3-ITD mutant-to-wild-type ratio > 0.5 with a concomitant *NPM1* mutation was associated with an intermediate risk, and the co-occurrence with wild-type *NPM1* was related to unfavorable risk. Currently, the updated 2022 ELN risk classification no longer considers the FLT3-ITD allelic ratio [75,76]. In the absence of adverse-risk genetic lesions listed in Table 2, the presence of FLT3-ITD is associated with intermediate risk irrespective of a concurrent *NPM1* mutation. Mutated *NPM1* is considered a favorable risk only in the absence of an FLT3-ITD mutation. The differences in the 2017 and 2022 ELN risk classifications are depicted in Table 2, highlighting the genetic markers used, including *FLT3* gene mutations. According to the meanwhile revised 2017 National Comprehensive Cancer Network (NCCN) guidelines, FLT3-ITD-mutated AML, even with normal cytogenetics, is linked with a poor risk [53,77]. Thus, both the ELN and NCCN guidelines emphasize FLT3-ITD as a negative prognostic marker and the importance of analyzing *FLT3* mutations within AML diagnostics. A Japanese multicentric prospective observational cohort study (Hokkaido Leukemia Net) recently demonstrated the inferior OS of FLT3-ITD-mutated AML patients compared to *FLT3*-mutation negative patients and highlighted FLT3-ITD as an independent poor prognostic factor [42]. In addition to its diagnostic significance, the determination of the *FLT3* mutation status and the assessment of the genetic landscape beyond are crucial for subsequent therapeutic decisions [78,79].

## 5. Biological and Clinical Consequences of FLT3-ITD Insertion Site Heterogeneity

### 5.1. Non-JMD ITDs Confer a High-Risk Phenotype in FLT3-ITD-Mutated AML

Unexpectedly, the investigation of ITD IS in 753 patients with unselected FLT3-ITD-positive AML showed that about 30% of ITDs display IS at the non-JMD level of FLT3, particularly within the β1-sheet of TKD1 [8,13]. ITDs were rarely identified within the nucleotide binding loop or the β2-sheet. This observation was confirmed by subsequent studies of ITD IS in large cohorts of AML patients [5,8]. The location of ITD IS with respect to the structural domains of the FLT3 receptor is highlighted in Figure 1.

Later on, additional independent studies confirmed that about 30% of FLT3-ITD mutations insert at non-JMD sites [5,7,14,80]. Schnittger et al. described a less favorable outcome in AML patients harboring FLT3-ITD mutations with more C-terminally located mutational start or end sites, thus including TKD1 [8]. Independent of the karyotype, a poorer median OS was observed in patients with more C-terminally located FLT3-ITD mutational start points [8]. In a multivariable analysis of a cohort of 241 younger adult patients aged 18–60 years with FLT3-ITD-mutated AML, a non-JMD FLT3-ITD located in the β1-sheet of the TKD1 was shown to be an unfavorable prognostic factor for achieving CR after induction therapy as well as for RFS and OS after allogeneic hematopoietic stem cell transplantation (HSCT) [5]. In a Chinese study from Liu et al. of 154 adult patients aged 18–65 years with FLT3-ITD-mutated AML, non-JMD ITDs, particularly within the β1-sheet of the TKD1, were associated with overall worse RFS at six months and OS at three years. All patients received standard induction and consolidation treatment; however, the impact of different postinduction therapies was not taken into account [7]. According to Schlenk et al., the unfavorable prognostic impact of non-JMD-inserting ITDs is independent of postinduction treatment after the first CR [14]. In 231 FLT3-ITD-mutated AML patients receiving postinduction therapy with either chemotherapy and autologous or allogeneic HSCT, a trend toward a higher cumulative incidence of relapse (CIR) for ITDs at the TKD1 level was described. Besides a high FLT3-ITD mutant-to-wild-type ratio > 0.51, FLT3-ITD IS at the TKD1 level was overall associated with worse CR rates, RFS, and OS. Interestingly, prognostic favorable effects of a concomitant *NPM1* mutation on achieving CR were related only to JMD-inserting ITDs [14]. 

In summary, a large body of evidence highlighted the negative prognostic impact of FLT3-ITD mutations inserted at non-JMD sites in various AML patient cohorts and at different times of treatment. Thus, in addition to the negative prognostic impact of an FLT3-ITD mutation on CR rates, OS, and RFS already acknowledged within current risk stratifications, ITDs inserted at the non-JMD level are associated with an even worse high-risk prognostic phenotype in FLT3-ITD-mutated AML.

### 5.2. Heterogeneity of TKI Response Depending on FLT3-ITD Insertion Sites—Preclinical and Clinical Studies

Interestingly, in vitro and in vivo preclinical models showed heterogeneous effects on TKI treatment depending on the particular localization of FLT3-ITD IS [26]. Arreba-Tutusaus et al. analyzed the in vitro response to midostaurin and quizartinib treatment of 32D and Ba/F3 cells transduced with constructs of FLT3-ITD exhibiting different IS. Compared to JMD ITDs, non-JMD ITDs inserted at the TKD1 level displayed lower sensitivity to midostaurin and quizartinib [26]. Similar results were obtained by Massacci et al.: Ba/F3 cells carrying FLT3-ITD with non-JMD IS were less susceptible to TKI treatment with midostaurin, quizartinib, and gilteritinib [15]. Furthermore, Arreba-Tutusaus et al. developed a retroviral transduction model in which two FLT3-ITD-mutated cell populations with ITD IS at the JMD and TKD1 level, respectively, were transplanted into irradiated recipient mice. In response to midostaurin as single-agent therapy for 10 days, a significant reduction of FLT3-ITD-mutated cells was observed only in the cell population exhibiting a JMD FLT3-ITD mutation, whereas non-JMD TKD1 inserting FLT3-ITD-mutated 32D cells remained largely unaffected [26]. Analyzing leukemic blast cells of an AML patient with an FLT3-ITD mutation and primary resistance to midostaurin, Breitenbuecher et al. identified the non-JMD-inserting FLT3-ITD variant A627E. TKI resistance to midostaurin was also demonstrated in vitro using hematopoietic 32D cells transfected with the FLT3-ITD A627E variant. In a syngeneic mouse model, injection of 32D cells with ITDs integrating between codons 598 and 599 or within a non-JMD IS (A627E) induced a lethal hematopoietic disease [81]. In summary, various preclinical in vitro and in vivo models showed TKI resistance in FLT3-ITD-mutated AML displaying non-JMD ITD IS.

The prospective randomized, placebo-controlled phase III CALGB 10603/RATIFY trial (NCT00651261) investigated the therapeutic benefit of midostaurin as a multi-targeted kinase inhibitor in combination with standard chemotherapy with cytarabine and daunorubicin in patients with newly diagnosed *FLT3*-mutated AML [44]. A previously conducted phase Ib study on a small number of overall 69 AML patients showed an improvement in CR rates in FLT3-ITD-mutated AML when midostaurin was added to standard chemotherapy, indicating promising therapeutic effects of TKI [82]. A total of 717 patients were randomized in the phase III CALGB 10603/RATIFY trial, which evaluated midostaurin treatment (360 patients) versus placebo (357 patients). In the group of midostaurin treatment, OS and event-free survival were significantly longer [43]. Thus, in 2017, the FDA approved midostaurin for the treatment of *FLT3*-mutated AML. A study by the German–Austrian AML Study Group showed improved event-free survival with midostaurin treatment in FLT3-ITD-positive AML patients aged 18–70 years [83]. In 2021, Rücker et al. performed an exploratory post hoc analysis of the CALGB 10603/RATIFY trial aiming to assess the molecular landscape of FLT3-ITD-mutated AML and to determine the prognostic impact of ITD IS in a total of 555 AML patients carrying FLT3-ITD [84]. Depending on ITD localization at either JMD or TKD1 level alone or with IS in both domains, three molecular subgroups were distinguished. In line with previous findings, non-JMD ITDs occurred in about 30% of FLT3-ITD-mutated AML patients [5,7,13,14,80]. By means of Cox regression, a multivariate analysis of OS revealed ITDs at the TKD1-sole level as a negative prognostic factor. Thus, FLT3-ITD-mutated AML patients with ITD IS in TKD1-sole had a significantly inferior 4-year OS compared with JMD-sole or JMD/TKD1 (*p* = 0.032). The 4-year OS rates were 29%, 44%, and 50%, respectively. In addition, there was a higher 4-year CIR in patients with ITD insertions in TKD1-sole compared to JMD-sole (60% vs. 45%). The rate of CR achievement was independent of FLT3-ITD IS [84]. Multivariate analysis for OS identified TKD1-sole IS, higher leukocyte counts, and older age as significant unfavorable factors [84]. The second most important clinical finding was that the beneficial effect of midostaurin on OS and CIR was restricted to the JMD-sole subgroup only. Thus, the 4-year rates for OS of patients on midostaurin or placebo were 48% vs. 40% (*p* = 0.047) for JMD-sole. The negative prognostic impact of FLT3-ITD inserting at the TKD1 level was not reversible via midostaurin treatment [84]. This finding is in line with the above-mentioned preclinical data showing resistance to midostaurin and other FLT3-TKIs in cells exhibiting ITDs with TKD1 IS [15,26]. Next-generation sequencing (NGS-cAR) was performed in the study by Rücker et al. to investigate the relationship between ITD IS and patient outcomes. NGS has been shown to be able to resolve clonal heterogeneity and sequence variability in the assessment of FLT3-ITD-mutated AML [85,86,87,88]; however, the ELN recommends using capillary electrophoresis for better determination of longer ITDs [75]. Another analysis of a small cohort of 54 FLT3-ITD-mutated AML patients treated with midostaurin revealed a comparable distribution of JMD and non-JMD ITDs at diagnosis and disease progression. Furthermore, the distribution of FLT3-ITD IS was similar regardless of the persistence or loss of FLT3-ITD at progression [89]. 

### 5.3. Characterizing the Molecular Basis of Chemotherapy and TKI Resistance in Non-JMD FLT3-ITD Insertion Sites

Regarding FLT3-ITD-mediated oncogenic signaling pathways, Fleischmann et al. investigated the susceptibility of various ITD IS at either the JMD or TKD1 level to inhibition of N-glycosylation using tunicamycin, heat shock protein 90 (HSP90) using 17-AAG, or histone deacetylation using valproic acid. After treatment with tunicamycin and incubation with 17-AAG, Annexin V-negative cells were significantly decreased regardless of ITD IS. However, Ba/F3 cells that harbor the FLT3-ITD variant G613E inserted at the TKD1 level showed higher sensitivity to 17-AAG. Within the Ba/F3 G613E variant, expression of ERK and STAT5 was decreased upon valproic acid treatment [90]. Various FLT3-ITD-mutated Ba/F3 cell lines transfected with concurrent point mutations of the TKD (F691L, Y842C, and D835V) have been shown to be resistant to quizartinib but more susceptible to HSP90 inhibition compared to quizartinib-sensitive cells [91]. Similar results were obtained in a 32D cell transfection model: Inhibition of HSP90 resulted in the degradation of FLT3-ITD mutated 32D cells with concurrent TKD point mutations associated with secondary drug resistance to FLT3-TKIs [92]. 

Upregulation of MCL-1 has been shown in different cell lines with varying FLT3-ITD IS at the JMD (MOLM-13, MV4-11) or TKD1 level (32D, A627E variant) compared to FLT3 wildtype (RS4-11) [33]. Upregulated MCL-1 at the protein level was also detected in a cytarabine-resistant cell line MV4-11-R harboring the FLT3-ITD mutation at the JMD level [93]. In the FLT3-ITD variant A627E inserting at the TKD1 level, Breitenbuecher et al. described upregulation of anti-apoptotic MCL-1 and showed sustained activation of ERK1/2 despite midostaurin-mediated inhibition of FLT3. However, inhibition of ERK1/2 in addition to midostaurin resulted only in a small increase in apoptosis in cells with TKD1-inserting ITDs without suppressing the MCL-1 level. Interestingly, decreasing the MCL-1 level using *MCL-1*-specific small interfering ribonucleic acid (siRNA) restored susceptibility to midostaurin [81]. Suppression of MCL-1 with flavopiridol induced apoptosis in a non-JMD FLT3-ITD expressing 32D cell line with an A627E mutation [33]. Thus, MCL-1 may serve as a promising target protein to overcome chemotherapy resistance [94]. Current preclinical studies of MCL-1 inhibition or downregulation of MCL-1 via inhibiting B-cell lymphoma 2 (Bcl-2) in FLT3-ITD-mutated AML are promising. However, due to the sole use of cell lines with a JMD-inserting FLT3-ITD mutation (MOLM-13, MV4-11), the discrimination of different ITD IS is lacking [95,96]. 

Massacci et al. examined mechanisms of TKI resistance in FLT3-ITD-mutated AML by combining high-sensitive transcriptome, phosphoproteome, and proteome analysis with literature-derived signaling networks using the SignalingProfiler computational module [15]. For this purpose, Ba/F3 cell lines reliably expressing FLT3-ITD with IS either at JMD or TKD1 level were treated using a panel of FLT3-TKIs. Phosphoproteome profiles enabled a clear stratification between FLT3 activation status and ITDs IS. The SignalingProfiler computational module predicted the active state of the WEE1 kinase and its opposite regulation in the Ba/F3 cell line with TKD1-ITDs. Cyclin-dependent kinase (CDK) 1 has been shown to be regulated by FLT3 involving WEE1 and to be a key upstream regulator of pro- and anti-apoptotic proteins. Activation of pro-survival proteins such as MCL-1 and Bcl-2 as well as an inhibitory interaction between WEE1 and CDK1 were specifically associated with ITD IS at the TKD1 level. Only in the Ba/F3 cells expressing JMD-ITDs, midostaurin-mediated cell-cycle arrest in the G1 phase resulted in a significant decrease in proliferation with a higher sensitivity to induce apoptosis. However, only in the Ba/F3 cell line harboring JMD-inserting ITDs, proteins involved in apoptosis were hyperphosphorylated after quizartinib treatment. WEE1-mediated protection against midostaurin-mediated cell-cycle arrest in cells expressing TKD1-ITDs could be overcome in the Ba/F3 cell line model and blast cells isolated from AML patients by selectively inhibiting WEE1. In untreated Ba/F3 cells harboring FLT3-ITD exhibiting a TKD1 IS, Pugliese et al. detected increased ROS accumulation and increased extent of deoxyribonucleic acid (DNA) damage as assessed by γ-H2AX compared with JMD ITDs [97]. The histone protein γ-H2AX is a sensitive marker for DNA double-strand breaks. Following irradiation with overall 3 Gray (Gy), 32D cells with TKD1-ITDs revealed fewer foci of γ-H2AX compared to JMD-inserting ITDs, indicating more effective DNA damage repair in these cells [26]. 

In conclusion, the current body of evidence supports the view that distinct conformational changes in the FLT3 receptor mediated by JMD- versus TKD1-inserting ITDs lead to differences in phosphorylation patterns of FLT3 downstream signaling molecules, which in turn alter the response to therapeutic agents [15,97,98]. It has been recently suggested that the ITD-induced conformational change of the FLT3 receptor favors atypical interaction with CSF2RB resulting in phosphorylation and activation of CSF2RB in FLT3-ITD-mutated cells [30]. Subsequent conformational and protein structure changes mediated by FLT3-ITD possibly alter the interaction between FLT3 and TKIs, leading to TKI resistance. For various FLT3-TKD point mutations (D835), it was demonstrated how major changes in FLT3 geometry alter the binding of TKIs [99,100]. Structural alterations may then lead to ineffective inhibition of FLT3 autophosphorylation by TKIs [101]. Furthermore, differences in phosphorylation of FLT3 downstream signaling molecules possibly alter their oncogenic potential. In Figure 2, oncogenic signaling networks in FLT3-ITD-mutated AML within ITDs inserted at non-JMD sites are illustrated.

### 5.4. Heterogeneity of Chemotherapy Response Depending on FLT3-ITD Insertion Sites

In first-line therapy, cytarabine is the backbone in the induction and consolidation therapy of AML patients, except in acute promyelocytic leukemia (APL) [75,102,103]. Following its approval, the combination of midostaurin with induction or consolidation chemotherapy became the standard of care in FLT3-ITD-mutated AML. Prior to the era of TKIs, Kayser et al. investigated the impact of ITD IS on chemotherapy response and survival. FLT3-ITD inserting at the TKD1 level, more specifically within the β1-sheet of TKD1, was associated with a worse CR rate after induction therapy, indicating chemotherapy resistance [5]. 

Pugliese et al. comprehensively examined the response to cytarabine depending on the ITD IS in vitro. The Ba/F3 and 32D cell lines, harboring FLT3-ITD either at the JMD or TKD1 level, were treated with cytarabine [97]. Cell lines harboring FLT3-ITD IS within TKD1 showed equally reduced sensitivity to cytarabine compared to ITDs inserted at the JMD level. In vitro cytarabine treatment resulted in less accumulation of DNA damage in FLT3-ITD-mutated Ba/F3 cells with ITDs within the TKD1. The levels of DNA damage response and DNA repair following cytarabine treatment were shown to be similarly independent of ITD IS. Cytarabine had a positive effect on cell-cycle and mitosis processes leading to the upregulation of cell-cycle kinases in Ba/F3 cells with FLT3-ITD inserted at the TKD1. When the SignalingProfiler computational module was applied to mass spectrometry (MS)-based phosphoproteomics performed in FLT3-ITD-mutated Ba/F3 cells, the authors found that cytarabine responsiveness is associated with the deregulation of the CDK2-CDK7 pathway and the subsequent accumulation of cells in the S phase. Deregulation of the CDK2-CDK7 pathway was in line with increased expression of CDK2 and cyclin H, as well as cell-cycle progression by activating CDK7/cyclin H. Furthermore, pharmacological suppression of CDK7 synergistically acted with cytarabine, restoring the sensitivity of Ba/F3 cells exhibiting FLT3-ITD IS at the TKD1 level [97]. Along this line, blast cells isolated from FLT3-ITD-mutated AML patients with TKD1 IS were less susceptible to cytarabine. Interestingly, AML blast cells expressing FLT3-ITD insertions both at JMD and TKD1 appeared more sensitive to cytarabine treatment, suggesting a dominant effect of the JMD IS of FLT3-ITD in chemotherapy response [97]. In Table 3, promising targets and compounds of therapeutic approaches to overcome resistance to chemotherapy and TKIs in non-JMD-inserting FLT3-ITD are illustrated. 

## 6. Conclusions

Following the discovery of FLT3-ITD mutations as molecular markers in AML patients, their prognostic value was recognized and integrated into risk stratification guidelines for AML used to date. Non-JMD-inserting FLT3-ITD are associated with inferior CR rates, PFS, and OS. In vitro and in vivo analyses revealed inferior susceptibility to TKI treatment and chemotherapy in AML expressing FLT3-ITD with IS at the TKD1 level. However, despite marked differences in survival and therapy resistance depending on ITD IS, previous risk stratifications do not consider the crucial role of non-JMD ITDs. Furthermore, current studies often do not distinguish between JMD- and non-JMD-inserting FLT3-ITD, limiting data density. To our knowledge, the expression of FLT3-ITD in relation to different ITD IS has also not been examined in pediatric cohorts. We propose recognizing non-JMD ITDs as an independent adverse prognostic factor. Thus, refined risk stratification guidelines highlighting the prognostic differences between JMD and non-JMD IS in FLT3-ITD-mutated AML are needed. Due to its considerable prognostic and therapeutic importance, the determination of FLT3-ITD IS should be a mandatory part of the daily diagnostic routine in AML. Overcoming TKI and chemotherapy resistance in FLT3-ITD-mutated AML, especially associated with ITDs inserted at the non-JMD level, could lead to promising genotype- and patient-specific treatment approaches.

## Figures and Tables

**Figure 1 cancers-15-02991-f001:**
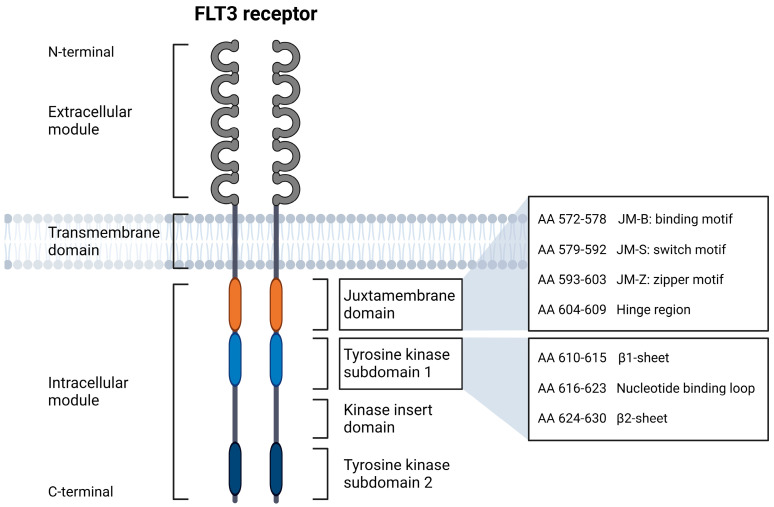
Structural domains of the FLT3 receptor in schematic representation, adapted from Breitenbuecher et al. [13] and Kayser et al. [5]. Created with BioRender.com (accessed on 25 May 2023).

**Figure 2 cancers-15-02991-f002:**
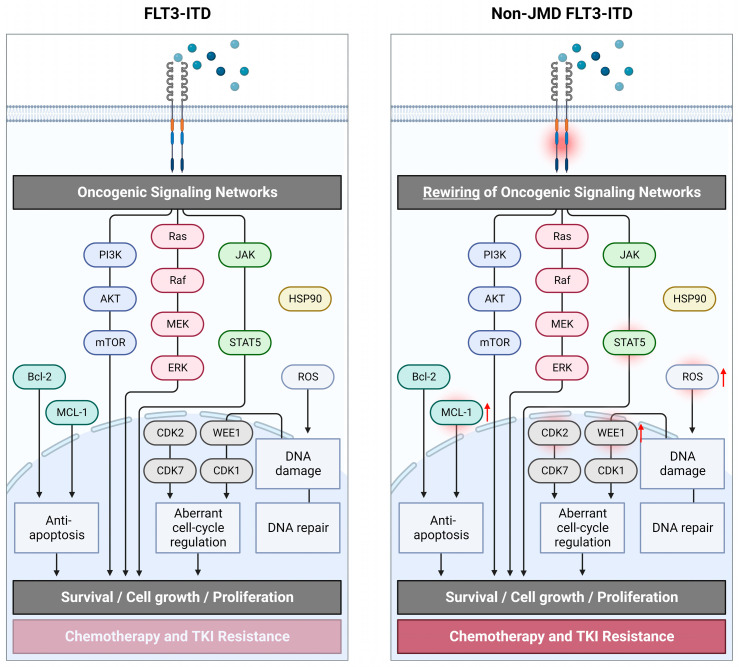
Schematic overview of oncogenic signaling in FLT3-ITD-mutated AML within FLT3-ITD inserting at non-juxtamembrane domain (non-JMD) sites. Rewiring of oncogenic signaling networks in non-JMD FLT3-ITD drives its oncogenic potential and resistance to chemotherapy and tyrosine kinase inhibitors. The overview of signaling pathways makes no claim of completeness. Created with BioRender.com (accessed on 25 April 2023).

**Table 1 cancers-15-02991-t001:** Overview of queries performed for the selection of references in accordance with the corresponding sections.

Query	Section
AML and FLT3-ITD (PubMed, last 5 years)	Introduction (Section 1)
	Signaling of FLT3 and FLT3-ITD (Section 2)
	FLT3 Tyrosine Kinase Inhibition (Section 3)
AML, FLT3-ITD, and Risk-Stratification (PubMed, last 5 years); AML, FLT3-ITD, and Prognostic Impact (PubMed, last 5 years)	Prognostic Impact of FLT3-ITD Mutations and Current Risk Stratification (Section 4)
AML, FLT3-ITD, and Insertion Site (PubMed); AML, FLT3-ITD, and TKD1 Mutation (PubMed)	Non-JMD ITDs Confer a High-Risk Phenotype in FLT3-ITD-Mutated AML (Section 5.1)
AML, FLT3-ITD, and RATIFY trial (PubMed); AML, FLT3-ITD, and TKD1 Mutation (PubMed)	Heterogeneity of TKI Response Depending on FLT3-ITD Insertion Sites—Preclinical and Clinical Studies (Section 5.2)
AML, FLT3-ITD, Signaling, and Domain (PubMed, last 5 years)	Characterizing the Molecular Basis of Chemotherapy and TKI Resistance in Non-JMD FLT3-ITD Insertion Sites (Section 5.3)
AML, FLT3-ITD, Chemotherapy Resistance, and Cytarabine (PubMed)	Heterogeneity of Chemotherapy Response Depending on FLT3-ITD Insertion Sites (Section 5.4)

**Table 2 cancers-15-02991-t002:** Acute myeloid leukemia (AML) 2017 European LeukemiaNet (ELN) and 2022 ELN risk classification by genetics at initial diagnosis (shortened version) [72,75]. Genetic markers, including *FLT3*, are highlighted with bold.

Risk Category	2017 ELN [72], Genetic Abnormality	2022 ELN [75], Genetic Abnormality
Favorable	- t(8;21)(q22;q22.1)/RUNX1-RUNX1T1 - inv(16)(p13.1;q22) or t(16;16)(p13.1;q22)/CBFB-MYH1 - **Mutated NPM1 without FLT3-ITD or with FLT3-ITDlow (<0.5)** - Biallelic mutated CEBPA	- t(8;21)(q22;q22.1)/RUNX1::RUNX1T1 - inv(16)(p13.1;q22) or t(16;16)(p13.1;q22)/CBFB::MYH1 - **Mutated NPM1 without FLT3-ITD** - bZIP in-frame mutated CEBPA
Intermediate	- **Mutated NPM1 and FLT3-ITDhigh (≥0.5)** - **Wild-type NPM1 without FLT3-ITD or with FLT3-ITDlow (<0.5) (without ad verse-risk genetic lesions)** - t(9;11)(p21.3;q23.3);MLLT3-KMT2A - Cytogenetic abnormalities not classified as favorable or adverse	- **Mutated NPM1 with FLT3-ITD** - **Wild-type NPM1 with FLT3-ITD (without adverse-risk genetic lesions)** - t(9;11)(p21.3;q23.3)/MLLT3::KMT2A - Cytogenetic and/or molecular abnormalties not classified as favorable or adverse
Adverse	- t(6;9)(p23;q34.1)/DEK-NUP214 - t(v;11q23.3)/KMT2A-rearranged - t(9;22)(q34.1;q11.2)/BCR-ABL1 - inv(3)(q21.3;q26.2) or t(3;3)(q21.3;q26.2)/GATA2, MECOM(EVI1) - −5 or del(5q); −7; −17/abn(17p) - Complex karyotype, monosomal karyotype - **Wild-type NPM1 and FLT3-ITDhigh (≥0.5)** - Mutated RUNX1 - Mutated ASXL1 - Mutated TP53	- t(6;9)(p23.3;q34.1)/DEK::NUP214 - t(v;11q23.3)/KMT2A-rearranged - t(9;22)(q34.1;q11.2)/BCR::ABL1 - t(8;16)(p11.2;p13.3)/ KAT6A::CREBBP - inv(3)(q21.3;q26.2) or t(3;3)(q21.3;q26.2)/GATA2, MECOM(EVI1) - t(3q26.2;v)/MECOM(EVI1)-rearranged - −5 or del(5q); −7; −17/abn(17p) - Complex karyotype, monosomal karyotype - Mutated ASXL1, BCOR, EZH2, RUNX1, SF3B1, SRSF2, STAG2, U2AF1, and/or ZRSR2 - Mutated TP53

**Table 3 cancers-15-02991-t003:** Overview of promising targets and compounds of therapeutic approaches to overcome resistance to chemotherapy and tyrosine kinase inhibitors in non-JMD-inserting FLT3-ITD. The list makes no claim of completeness.

Target	Compound	Combination	Model	Reference
CDK7	THZ1	Cytarabine	Preclinical (Ba/F3 cells, patient-derived blast cells)	Pugliese et al. [97]
HSP90	17-AAG	Tunicamycin	Preclinical (Ba/F3 cells)	Fleischmann et al. [90]
MCL-1	*MCL-1*-specific siRNA	Midostaurin	Preclinical (32D cells)	Breitenbuecher et al. [81]
	Flavopiridol	ABT-737	Preclinical (32D cells)	Kasper et al. [33]
WEE-1	Adavosertib (MK1775)	Midostaurin	Preclinical (Ba/F3 cells, patient-derived blast cells)	Massacci et al. [15]

## Data Availability

Data sharing not applicable.

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
