# Peer review of "How ITD Insertion Sites Orchestrate the Biology and Disease of FLT3-ITD-Mutated Acute Myeloid Leukemia"

_cancers, 2023, doi:10.3390/cancers15112991_

Round 1
Reviewer 1 Report
This manuscript describes a subgroup of AML harboring FLT3ITD mutation in non-canonical JMD, being associated with worse clinical outcome and resistance to both chemotherapy and tyrosine kinase inhibition. Recent works hypothesized a different pathway activation in the non-JMD FLT3-ITD, suggesting new therapeutic approaches that could ameliorate clinical response.
Overall, this manuscript represents an interesting and timely analysis with clinical and therapeutical implication.
Some minor suggestions:
Regarding 5.1 and 5.2, It would be useful a table summarizing outcome of FLT3ITD at non-JMD sites with respect to JMD, and TKI response in different cohorts analysed up to now.
In chapter 5.3 it is not clear to me whether MCL-1 upregulation occur exclusively in FLT3ITD non-JMD; authors reported MCL-1 upregulation also in MV4;11 that harbors FLT3ITD in the JMD domain (https://ashpublications.org/blood/article/122/21/1382/13799/FLT3-Variant-Detection-In-Cell-Lines). I think that it would be useful to clarify this point, according to the last paragraph sentence that states “Current preclinical studies of MCL-1 inhibition or downregulation of MCL-1 by inhibiting B-cell lymphoma 2 (Bcl-2) in FLT3-ITD-mutated AML are promising but often lack analysis regarding different ITD IS.” .
Moreover, Table 3 shows therapeutic approaches to overcome resistance to CT and TKI in non_JMD inserting FLT3ITD, but the models described in the table mentioned MV4;11 and MOLM13, which do have FLT3ITD in their JMD. Please modify or clarify accordingly.
To be thorough, authors might report if FLT3ITD non-JMD has been studied in pediatric cohorts.
Author Response
We would really like to thank the reviewer for the thorough and constructive review to improve our manuscript. Please see the attachment.

Reviewer 2 Report
Mutations in protein tyrosine kinases occur either in the kinase domain or in regulatory domains. In Receptors such as FLT3 or KIT, the regulatory domain mutants are very sensitive to kinase inhibitors while mutations in the catalytic domain are often resistant to class II inhibitors.
FLT3 ITD mutations, one of the most common mutations found in AML, were defined as duplications in the regulatory juxtamembrane domain of the receptor. However, closer analyses of these mutations have revealed a more complex architecture with about 1/3 duplications that also affect the TK1 lobe of the catalytic domain, with functional consequences on the FLT3 oncoprotein. The manuscript is a review that recapitulates and discusses the accumulated knowledge on the diversity of ITD mutations, and the specific features of juxtamembrane ITD vs TK1 ITD. Overall, the manuscript is of great interest as it summarizes the data accumulated on this very important topic, that has been overlooked while it has profound implications to understand the disease and for the patients (prognosis factor and resistance to therapy). The manuscript is clearly written, balanced and highlights important perspectives.
The following points need corrections or clarifications.
11- Lane 53: FLT3 does not have two tyrosine kinase domains. In FLT3 (and all the member of the PDGFR family), the catalytic kinase domain is split in two by a kinase insert sequence. The two parts were called TK1 (the ATB-binding lobe) and TK2 (the phopho-transferase region). The bottom line is that there is only one catalytic tyrosine kinase domain. Please correct lane 53 and Figure 1, where “tyrosine kinase domain 1 and 2” should be “subdomain 1 and 2” or something equivalent (ATP binding lobe and phopho-transferase lobe, for instance).
22- In my opinion Table 1 is not useful.
33- Lanes 128-138, discussion on resistance mechanisms. One major reason for resistance and relapse is tumor heterogeneity; in the context of AML with FLT3 mutations, the selection of minor clones that do not have the mutation is well documented as an escape mechanism. Please include clonal heterogeneity in this chapter on resistance.
44- Lanes 145-149: FLT3-ITD mutations is a poor prognosis factor in AML. This is well documented in many published papers. Please cite the first studies instead of reference 42 which is a manuscript published in 2023.
5- Part 5.3 Molecular basis of TKI resistance.
A main factor for TKI resistance in most kinases, including FLT3 point variants, are protein structure changes. The mutations do not allow access of some small molecule inhibitors into the ATP pocket of the catalytic domain (as it is the case for FLT3 D835 mutation). This is not discussed in the chapter. The analysis of FLT3 kinase domain structure in the context of ITD-TKD1 would be very informative on the question of kinase inhibitor resistance. Please include a sentence or few sentences mentioning tyrosine domain structure alterations by mutations.
Author Response

(The authors gave the same response as above.)
